# Negative Affectivity and Emotion Dysregulation as Mediators between ADHD and Disordered Eating: A Systematic Review

**DOI:** 10.3390/nu12113292

**Published:** 2020-10-27

**Authors:** Sarah El Archi, Samuele Cortese, Nicolas Ballon, Christian Réveillère, Arnaud De Luca, Servane Barrault, Paul Brunault

**Affiliations:** 1Qualipsy, EE 1901, Université de Tours, 37041 Tours, France; sarah.elarchi@univ-tours.fr (S.E.A.); christian.reveillere@univ-tours.fr (C.R.); servane.barrault@univ-tours.fr (S.B.); 2Center for Innovation in Mental Health, Academic Unit of Psychology, Clinical and Experimental Sciences (CNS and Psychiatry), Faculty of Medicine, University of Southampton, Southampton SO17 1BJ, UK; samuele.cortese@gmail.com; 3Solent NHS Trust, Southampton SO19 8BR, UK; 4New York University Child Study Center, New York, NY 10016, USA; 5Division of Psychiatry and Applied Psychology, School of Medicine, University of Nottingham, Nottingham NG72UH, UK; 6CHRU de Tours, Service d’Addictologie Universitaire, Équipe de Liaison et de Soins en Addictologie, 37044 Tours, France; nicolas.ballon@univ-tours.fr; 7UMR 1253, iBrain, Université de Tours, INSERM, 37032 Tours, France; 8CHRU de Tours, Centre Spécialisé de l’Obésité, 37044 Tours, France; a.deluca@chu-tours.fr; 9UMR 1069, Nutrition, Croissance et Cancer, Université de Tours, INSERM, 37032 Tours, France; 10CHRU de Tours, Service d’Addictologie Universitaire, Centre de Soins d’Accompagnement et de Prévention en Addictologie d’Indre-et-Loire (CSAPA-37), 37000 Tours, France

**Keywords:** food addiction, addictive-like eating, binge eating, eating disorders, loss of control overeating, Attention-Deficit/Hyperactivity Disorder, emotion self-regulation, negative mood

## Abstract

Attention-Deficit/Hyperactivity Disorder (ADHD) is associated with disordered eating, especially addictive-like eating behavior (i.e., binge eating, food addiction, loss of control overeating). The exact mechanisms underlying this association are unclear. ADHD and addictive-like eating behavior are both associated with negative affectivity and emotion dysregulation, which we hypothesized are mediators of this relationship. The purpose of this systematic review was to review the evidence related to this hypothesis from studies assessing the relationship between childhood or adulthood ADHD symptomatology, negative affectivity, emotion dysregulation and addictive-like eating behavior. The systematic review followed the Preferred Reporting Items for Systematic reviews and Meta-Analyses (PRISMA) recommendations. The literature search was conducted in PubMed and PsycINFO (publication date: January 2015 to August 2020; date of search: 2 September 2020). Out of 403 potentially relevant articles, 41 were retained; 38 publications reported that ADHD and disordered eating or addictive-like eating behavior were significantly associated, including 8 articles that suggested a mediator role of negative affectivity or emotion dysregulation. Sixteen publications reported that the association between ADHD symptomatology and disordered eating or addictive-like eating behavior differed according to gender, eating behavior and ADHD symptoms (hyperactivity, impulsivity and inattention). We discuss the practical implications of these findings and directions future research.

## 1. Introduction

Attention-Deficit/Hyperactivity Disorder (ADHD) is a neurodevelopmental disorder characterized by impairing levels of inattention and/or hyperactivity-impulsivity, which is thought to begin generally in childhood (before the age of 12) and significantly interferes with social, academic, and/or occupational functioning. Childhood ADHD prevalence is estimated to be between 5 and 7% [1,2,3]. Current evidence indicates that impairing symptoms of the disorder persist in adulthood in 50 to 60% of cases [4]. The prevalence of adult ADHD is between 1.4 and 3.6% [1]. The treatment for individuals with ADHD includes pharmacologic [5] and non-pharmacologic [6] options. It has been demonstrated that both childhood and adult ADHD is associated with higher prevalence and risk of a large number of medical and psychiatric comorbidities. According to Kooij and colleagues (2019), 60–80% of individuals with ADHD show life-time comorbidities such as anxiety disorder (34%), mood disorder (22%), behavioral disorder (15%) and substance use disorders (11%). One of the most prevalent medical comorbidities is obesity; meta-analytic evidence indicates a 70% increased risk of obesity in adults with ADHD compared to those without ADHD [7,8]. ADHD has also been found to be significantly associated with eating disorders (EDs) (i.e., anorexia nervosa [AN], bulimia nervosa [BN], and binge eating disorder [BED]) [9]. In addition, ADHD is associated more generally with addictive-like eating behavior, even when no ED is diagnosed, notably loss of control overeating [10], binge eating (i.e., recurrent consumption of unusually large amounts of food during a discrete period of time while experiencing loss of control over food intake), and food addiction (FA) (i.e., addictive-like eating behaviors in relation to specific foods high in fat and/or refined carbohydrates, including craving, loss of control overeating, harm related to the behavior, and maintenance of the behavior despite negative consequences) [11,12,13].

An important research area related to addictive-like eating behavior focuses on the “food addiction” phenotype. According to Gearhardt and colleagues (2009) [12], this can be measured by applying the Diagnostic and Statistical Manual of mental disorders (DSM) criteria for substance dependence to highly palatable foods. FA has been assessed in the general population, among individuals with obesity or ED [14], and with impulse control disorders and psychiatric disorders, including major depressive disorder [15], substance use disorders [16], post-traumatic stress disorder [17], and ADHD [11]. Although FA is not part of the DSM-5 [18] and remains a hotly debated topic, a growing body of literature demonstrates that the “food addiction” phenotype shares some risk factors with other addictive behaviors and could improve our understanding of disordered eating behavior. On the one hand, FA shares neurobiological and clinical features with substance use disorder, such as reward system involvement, loss of control over intake, experience of craving and high impulsivity. On the other hand, it shares features with binging type ED, such as eating a large amount of food in a discrete period of time, and a sense of lack of control overeating during this episode [18]. In fact, FA is over-represented among EDs, especially the binging/overeating types (BN, BED and binging subtype AN [19,20]), but can also be present when no ED is diagnosed. According to Maxwell, Gardiner and Loxton (2020), FA and binge eating are associated with impulsivity, and “there seems to be a pattern emerging regarding overconsumption of food, task effort and lack of inhibition control, specifically that FA is associated with an inability to put the “brakes” on behavior” [21].

Different explanations have been proposed to explain the association between adult ADHD and addictive-like eating behavior. One hypothesis is that the impulsivity dimension of ADHD symptoms may explain the co-occurrence of ADHD and addictive-like eating behavior, such as binge eating [22]. The impulsivity associated with ADHD may increase the overall risk of sensation seeking and addictive disorders, including both substance-use disorders and behavioral addictions [23,24]. Urgency, defined as the tendency to commit rash or regrettable actions as a result of intense negative affect [25], has been hypothesized to be one of the main facets of impulsivity explaining the association between ADHD and addictive disorders [26,27]. As reported by Van Emmerik-Van Oortmerssen and colleagues (2012) in their meta-analysis [28], 23.1% of individuals with a substance-use disorder meet DSM criteria for ADHD. In addition, Anker, Bendiksen and Heir (2018) found that the prevalence of substance-use disorder among the ADHD population ranged from 4% to 23.6%, depending on gender or the substance used [29]. Similarly, addictive disorders are over-represented among people with ADHD [30,31,32]. Some publications [31,32,33] report that inattention and hyperactivity/impulsivity are related to the severity of addictive behavior, notably in gambling disorder and symptoms of internet addiction as assessed by the Internet Addiction Test [34]. They also posit that emotion self-regulation may be an important mediator in the association between ADHD and addictive disorders, highlighting the need for a systematic review in this field.

Another hypothesis regarding the relationship between adult ADHD and disordered eating behavior concerns the emotional self-regulation difficulties observed in both groups. Emotion regulation refers to conscious and unconscious processes regulating emotions. “Because emotions are multicomponential processes that unfold overtime, emotion regulation involves changes in emotion dynamics, or the latency, rise time, magnitude, duration, and offset of responses in behavioral, experiential, or physical domains” [35]. Five types of emotion regulation strategies have been described: situation selection (selecting situations that avoid uncomfortable emotions), situation modification (modifying situation features that lead to uncomfortable emotions), attentional deployment (distracting oneself from the attention-grabbing features of an emotional situation), cognitive change (reappraising the emotional meaning of a situation in non-emotional terms) and response modulation (modulating the behavioral, experiential, or physical aspect of the emotional response) [36]. Disruption of these processes leads to difficulties in generating and controlling emotions, associated with inappropriate behavior. Emotion regulation difficulties are encountered in some disorders, including ADHD [37], substance-use disorder [38] and disordered eating [39]. Masi and colleagues (2020) found that emotional dysregulation was a predictor of the persistence of ADHD symptoms after 4 weeks of pharmacological treatment. Higher levels of emotional dysregulation at the baseline assessment predicted higher levels of overall symptoms of ADHD at follow-up [40].

The hypothesis of a mediating role of emotion dysregulation in the association between ADHD and disordered eating is supported by the strong association found between emotion dysregulation and ED [39]. Emotion dysregulation affects up to 70% of adults with ADHD and substantially worsens the psychosocial outcomes of the disorder [41]. Moreover, the DSM-5 highlights emotion dysregulation as a feature supporting the diagnosis of ADHD [18]. According to the systematic review of ADHD-associated emotion dysregulation conducted by Beheshti, Chavanon and Christiansen (2020), the persistence of ADHD inattention symptoms in older age correlates with impaired situation identification, which requires attention processes, whereas hyperactive symptoms are associated more with impaired capacity to inhibit emotional responses. Additionally, emotional lability and negative emotional responses might play a key role in the emotion dysregulation-associated psychopathology of adults with ADHD [42]. Emotion dysregulation has been identified as a mediator between ADHD symptoms and several disorders such as depressive symptoms [43]. Emotion regulation difficulties particularly concern negative affect. Negative affectivity has been shown to be higher in individuals with ADHD and to be associated with a negative impact on ADHD experience and medication adherence, and increased risk of suicidal ideation and behavior, or various comorbid disorders [44,45,46]. Individuals with ADHD also show lack of emotion regulation strategies. As hypothesized for persons with a substance-use disorder [47], individuals who are less likely to use coping strategies to deal with or express emotions may resort to more problematic behavior. We can hypothesize that substance-use disorder and addictive disorders may provide immediate pleasure and/or a dissociative-like state to individuals with ADHD, offering psychological escape from the offending reality [48], and thus constitute a dysfunctional coping strategy to regulate negative affect.

The role of emotion dysregulation in the association between ADHD and addictive behavior has also been investigated in gambling disorder. For example, Mestre-Bach and colleagues (2019) found people with gambling disorder and ADHD symptomatology had greater emotion regulation difficulties than those without ADHD. The authors found that individuals with ADHD-gambling disorder comorbidity had higher rates of the following emotion regulation difficulties: non-acceptance of emotional responses, difficulty pursuing goal-directed behaviors when experiencing negative emotions, difficulty controlling impulsive behaviors when experiencing negative emotions, limited access to emotion regulation strategies, and lack of emotional clarity [32]. Their results are in line with the mediating role of emotion regulation in the relationship between ADHD symptomatology and addictive disorders in patients with gambling disorder. However, to our knowledge, no systematic review has been conducted to assess the mediating role of emotion regulation in ADHD symptoms and ED/addictive-like eating behavior (i.e., FA, binge eating, loss of control overeating).

To fill this gap, the aim of this study was to conduct a systematic review of studies investigating the association between childhood/adult ADHD, negative affectivity, emotion regulation, and disordered eating, with a specific focus on addictive-like eating behavior (i.e., binge eating, FA, loss of control overeating). We investigated negative affectivity, a common term involving many negative emotions such as anxiety, depression, negative urgency, stress. To this end, we first explored the characteristics of studies conducted in this field of research. In order to investigate the association between ADHD and disordered eating, we examined the prevalence of ADHD and disordered eating comorbidity within different populations. Next, we assessed negative affectivity and emotion regulation in individuals with ADHD, and finally we examined the involvement of these features in the relationship between ADHD symptomatology and addictive-like eating behavior. Due to potential difference in these relationships between children/adolescents and adults, we investigated both populations.

We hypothesized that: (1) individuals with disordered eating would show more ADHD symptoms; (2) individuals with ADHD symptoms would have higher levels of disordered eating; (3) ADHD symptoms would be associated with severity of addictive-like eating behavior; (4) the level of ADHD symptoms would be associated with high levels of negative affectivity and emotion regulation difficulties; (5) negative affectivity and emotion regulation difficulties may be mediators in the relationship between ADHD symptoms and addictive-like eating behavior.

## 2. Materials and Methods

This systematic review included publications investigating the association between ADHD and addictive-like eating behavior such as loss of control overeating, binge eating, and preoccupation with food, which are the main FA symptoms, and some DSM-5 EDs (eating disorders mentioned in the Diagnostic and Statistical Manual of Mental Disorders, Fifth edition) such as BN and BED, which show high FA prevalence [19,20].

This review was undertaken according to the quality standards of the Preferred Reporting Items for Systematic reviews and Meta-Analyses (PRISMA; Figure 1).

### 2.1. Literature Search

We conducted the literature search on 2 September 2020; the systematic literature review methodology included analysis of the electronic databases PsycINFO and PubMed. In order to identify all relevant publications on the association between [ADHD] and [FA symptoms and/or disordered eating], we used the following key words: [“ADHD” OR “attention-deficit hyperactivity disorder”] AND [“food addiction” OR “binge eating” OR “eating disorder” OR “bulimia” OR “obesity” OR “obese” OR “overweight”]. We included studies that used these keywords in their abstract (criterion I1, see Table 1). We focused on articles published from January 2015 to August 2020 (criterion I2) in peer-reviewed journals (criterion I3). Moreover, as we did not have funding for translation, we only included publications written in English or French (criterion I4). Based on these inclusion criteria, we excluded book chapters, letters to the editor and articles published before January 2015 and not written in English or French (criteria E1–E4). After removing duplicates, 403 article abstracts were identified for “abstract screening”.

Careful reading of these abstracts allowed us to select articles with an empirical approach (criterion I5), concerned directly or indirectly with ADHD and eating behavior (criterion I6) and investigating ADHD and disordered eating symptoms in the same individual (criterion I7). These inclusion criteria led to exclusion of review and meta-analysis articles (criterion E5), publications which did not address ADHD and eating behavior directly or indirectly, or focused on ADHD treatment or medical imaging (criterion E6). We also excluded all publications that investigated the impact of parents’ disordered eating or body mass index (BMI) on their child’s ADHD symptoms (criterion E7).

The papers thus retained were then read in full and appraised. We did not use a specific tool to appraise the quality of these studies, but they were checked for all the inclusion criteria and selection errors. We also checked that all the studies assessed ADHD and eating behavior using a validated instrument such as self-administered questionnaires or clinical interviews (criterion I8).

Regarding the characteristics of the populations studied, as our aim was to provide an overview of the association between ADHD and disordered eating, we did not consider age or gender as exclusion criteria.

### 2.2. Data Extraction

To investigate the characteristics of the publications, the following data were extracted: author names, country and year of publication, source, sample characteristics (age, gender, size, recruitment method and place), study design. We also extracted data about the prevalence of ADHD in individuals with disordered eating and the prevalence of disordered eating in individuals with ADHD. We thus identified the ADHD assessment tools used, the use of medication especially for individuals with ADHD, the type of eating behavior and the tools used to assess it. Finally, we examined the main results and conclusions about disordered eating and ADHD comorbidity. In this way, we extracted data regarding the association between ADHD and disordered eating, especially addictive-like eating symptoms and the involvement of negative affectivity and emotion self-regulation.

It should be noted that we use the word “symptom” to describe features of disordered eating and ADHD assessed only through self-administered questionnaires, and “diagnosis” or “severity” when assessment was through clinical interviews. Moreover, we use the word “eating disorder” (or ED) only for DSM disorders such as BN, BED and AN, and the word “disordered eating” as a generic word to include all pathological eating behaviors/symptoms such as binge eating, food addiction, loss of control overeating, strong desire for food, preoccupation with food, bulimic symptoms….

## 3. Results

We initially identified 403 articles, of which 97 were screened and selected for full-text reading. After full-text reading, 56 publications were excluded for the following reasons:-No data about behavioral features of eating (*n* = 38), including 30 publications which focused on the association between ADHD and BMI [8,49,50,51,52,53,54,55,56,57,58,59,60,61,62,63,64,65,66,67,68,69,70,71,72,73,74,75,76,77,78,79,80,81,82,83,84,85]-Non-representative sample, e.g., autism spectrum disorder (*n* = 4) [86,87,88,89]-Previous selection errors (*n* = 10) [22,90,91,92,93,94,95,96,97,98]-Investigations did not include ADHD-disordered eating association (*n* = 3) [99,100,101]-No access to full text (*n* = 1) [102]

Thus, 41 publications were included in this systematic literature review for qualitative synthesis (see Figure 1 for the study flow chart).

### 3.1. Article Characteristics

#### 3.1.1. Country of Investigation

The majority of these studies were conducted in the USA (*n* = 10, 25.6% of the included publications). Others were conducted in Sweden (*n* = 5, 12.8%), France, Canada (*n* = 4, 10.3% for each), the UK (*n* = 3, 7.7%), Spain, Brazil (*n* = 2, 5.1% for each), Norway, Australia, Israel, Korea, Switzerland, Greece, Iran, Germany and China (*n* = 1, 2.6% for each). One study did not specify the country of recruitment.

#### 3.1.2. Year of Publication

Included articles were published between January 2015 and August 2020. Eleven articles were published in 2017, 10 before 2017 and 20 after 2017. See Figure 2.

#### 3.1.3. Study Design

Among the 41 publications, 80.5% were cross-sectional (*n* = 33), and 19.5% were prospective longitudinal studies (*n* = 8).

#### 3.1.4. Age of Interest

Nineteen studies were conducted with children and/or adolescents (46.3%) and 24 with adults (58.5%). Two studies had a mixed adolescent-adult sample (4.9%).

#### 3.1.5. Population

Twenty-two studies were conducted with participants from the general population (53.7%), and 46.3% (*n* = 19) involved clinical populations: patients with severe obesity recruited in obesity and centers or prior to bariatric surgery (*n* = 8), patients with disordered eating (*n* = 6), ADHD outpatients (*n* = 2), or patients recruited in psychiatric departments (*n* = 3).

#### 3.1.6. ADHD Assessment and Medication

ADHD was assessed through clinical interviews (including semi-structured interviews) in 21 studies (51.2%), and through self-administered questionnaires in 20 studies (48.8%).

For children and adolescents, the main assessment tool for ADHD was the Kiddie Schedule for Affective Disorders and Schizophrenia (KSADS; 26.3% of the 19 studies conducted with children or adolescents) and the ADHD Rating Scale (ADHD-RS; 15.8%). For adults, ADHD was mainly assessed with DSM-IV or DSM5 semi-structured interviews using the Composite International Diagnostic Interview (CIDI), the Diagnostisch Interview Voor ADHD bij volwassenen (DIVA 2.0), or the Structured Clinical Interview for DSM Disorders (SCID). The main self-administered questionnaire was the Adult ADHD Self-Report Scale (ASRS; 41.7% of the 24 studies conducted with adults). It should be noted that some studies used the ASRS, a screening scale, as a diagnostic tool.

Fifty-four percent of the studies with adults included a retrospective assessment of childhood ADHD symptoms (*n* = 13), included in the diagnostic tool or additionally reported mainly through the Wender Utah Rating Scale (WURS) (*n* = 3).

Despite the known influence of ADHD pharmacological treatment on eating behavior [103], only 10 studies specified the ADHD medication status (25.6%). Three of them were conducted in medication-naïve populations, the remainder reported the rate of ADHD participants on medication.

#### 3.1.7. Disordered Eating Assessment Tools

Among the studies of children-adolescents, 7 (36.8%) assessed eating behavior through interviews (including semi-structured interviews), 10 (52.6%) through self-administered questionnaire, and 2 (10.5%) used both interviews and self-administered questionnaires. Various tools were used to assess disordered eating behavior, including the following self-administered questionnaires: the Eating Disorder Inventory-2 (EDI-2) (*n* = 3), the Children’s Eating Attitude Test (ChEAT), the Child Eating Behavior Questionnaire (CEBQ), the Eating Disorder Examination Questionnaire (EDE-Q) (*n* = 3 for each), and the Child Eating Disorder Examination (ChEDE), which specifically assesses loss of control overeating (*n* = 3). None of the studies used the Yale Food Addiction Scale for Children.

For adults, 14 studies (58.3%) used professional interviews (including semi-structured interviews), 14 publications (58.3%) were based on self-administered questionnaires investigating disordered eating, and 4 (16.7%) assessed disordered eating through both interviews and self-administered questionnaires. The main ED diagnostic tools used during clinical interview were the Mini International Neuropsychiatric Interview (MINI) and the SCID (*n* = 3 for each). The main self-administered questionnaires were the Binge Eating Scale (BES) to assess binge eating (*n* = 5), the original (DSM-IV-TR based) Yale Food Addiction Scale (YFAS) and the YFAS 2.0 (DSM-5 based) to assess FA (*n* = 2), the EDE-Q and the EDI-2 to assess disordered eating (*n* = 4 for each), and the Bulimic Investigatory Test Edinburgh (BITE) to assess bulimic symptoms (*n* = 4).

### 3.2. Association between ADHD and Disordered Eating

#### 3.2.1. Prevalence of Disordered Eating in Individuals with ADHD

##### Children and Adolescents

Four studies focused on the association between disordered eating and addictive-like eating behavior among children with ADHD symptoms (Table 2). They showed divergent results depending on the type of population. Wentz and colleagues (2019) [104], who assessed children recruited in an obesity clinic found no significant difference between individuals with and without ADHD diagnosis in terms of loss of control overeating. However, a study conducted in the general non-clinical population found a higher prevalence of loss of control overeating in children with than without ADHD diagnosis (70.5% vs. 20%; *p* < 0.001). The odds of loss of control overeating were increased 12.68 times for children with ADHD (95% Confidence Interval (CI): 3.11–51.64; *p* < 0.001) after adjusting for age, sex and race [105]. Another study with children attending psychiatric outpatient clinics found a higher prevalence of binge eating in individuals with ADHD than in controls (26% vs. 2%; *p* < 0.001) [103]. Moreover, in a longitudinal study by Bisset and colleagues (2019) [106], adolescents who screened positive for ADHD symptoms at age 12–13 tended to have a higher risk of objective binge eating at age 14–15 than adolescents without ADHD symptomatology (3.7% vs. 1.3%; Odds Ratio (OR) = 2.9, 95% CI: 0.9–8.6). Interestingly, this association was significant only for boys (2.9% vs. 0.3%; OR = 9.4, 95% CI: 1.7–52.8) and not for girls (6.5% vs. 2.2%; OR = 3.1, 95% CI: 0.7–14.0). The authors found no difference in terms of BN and BED symptoms (even partial syndromes) between adolescents with and without ADHD symptoms.

##### Adults

Within adult population, eight studies assessing disordered eating prevalence among individuals with ADHD symptomatology.

Two of these studies, with no control group, found a prevalence of 8.6% for BN [111], and 1.1% and 13% for any ED in ADHD patient men and women respectively [29]. Four studies with a general non-clinical population examined ED prevalence; ADHD-ED association odds ratio ranged from 1.32 (95% CI: 0.82–2.13) to 28.24 (95% CI: 6.33–126.01) [13,107,108,109]. These associations were particularly strong for BN (up to OR = 28.24, 95% CI: 6.33–126.01) [107,109]. Three of these studies found that ADHD symptoms were associated with an increased risk of ED. However, the odds ratio was significant after adjusting for age, sex and race, but not after adjusting for age, sex, race and psychiatric comorbidities, especially for BED (details in Table 2) [108,109]. Among psychiatric outpatients, Gorlin and colleagues (2016) [110] found higher ED prevalence for individuals diagnosed with ADHD (9.3% vs. 3.8%, *p* < 0.01), especially for the inattentive subtype (inattentive subtype: 10.3% individuals with an ED; OR = 3.01, 95% CI: 1.30–6.34; combined subtype: 8.1%, OR = 2.17, 95% CI: 0.90–4.68).

All publications assessing addictive-like eating symptoms in individuals with ADHD symptoms (*n* = 4) reported that ADHD was associated with a higher risk of addictive-like eating symptomatology: food addiction, binge eating, uncontrolled eating, significant distress in relation to food, and made him/herself be sick because he/she felt uncomfortably full [11,13,107,108] (details in Table 2). The FA prevalence rate was higher in patients with ADHD symptoms or diagnosis. In a study conducted in a non-clinical student population, FA prevalence was observed in 14.1% of the sample with ADHD symptoms compared to only 4% of those without ADHD symptoms (OR = 2.27, 95% CI: 1.05–4.88) [13]. In a sample of patients with severe obesity, FA prevalence was higher in those with than without ADHD diagnosis (28.6% vs. 9.1%; OR = 4.00, 95% CI: 1.29–12.40) [11]. Moreover, in a sample of adults with severe obesity, FA was associated with a retrospective assessment of childhood ADHD (24.3% vs. 8.8% without childhood ADHD symptoms, OR: 3.32, 95% CI: 1.08–10.23, *p* = 0.034) [11].

#### 3.2.2. Prevalence of ADHD in Individuals with Disordered Eating

##### Children and Adolescents

Three studies of overweight or obese children assessed ADHD prevalence (Table 3). One study with a non-clinical sample by Gowey and colleagues (2017) [112] found a rate of clinical levels of ADHD of 5% and subclinical levels of 5.91%, similar to the prevalence in the normal weight population. However, other studies conducted in clinical populations of children with obesity found higher rates of ADHD, ranging from 11% [113] to 18.4% [104]. Reinblatt and colleagues (2015) [105] found that the odds of children with obesity and loss of control overeating having an ADHD diagnosis was 7.3 times higher (95% CI: 1.88–28.17) than obese children without loss of control overeating, and 10.44 times higher (95% CI: 2.96–36.75) than children without obesity. These results were observed for both inattentive and hyperactivity/impulsivity ADHD subtypes.

Rojo-Moreno and colleagues (2015) [114] and Mohammadi and colleagues (2019) [115] assessed ADHD and eating disorder in general non-clinical populations. They found higher rates of ADHD in children with than without eating disorders ([114]: 31.4% vs. 8.4%, *p* < 0.05; [115]: 7.6% vs. 3.9%, *p* = 0.026). Furthermore, Kim and colleagues (2018) [116] found that 21.1% of children presenting with addictive-like eating behavior such as every-day overeating had a high risk of ADHD (see Table 4).

##### Adults

Three studies conducted in adults with severe obesity, recruited in obesity hospital departments, reported the prevalence of ADHD (Table 3). Nielsen and colleagues (2017) [117] estimated that 8.3% of bariatric surgery patients screened positive for ADHD on both the WURS (childhood ADHD symptoms scale) and the CAARS (adult ADHD symptoms scale). Based on adult ADHD DSM-IV criteria (including ADHD symptoms before the age of seven years), Brunault and colleagues (2019) [11] and Nazar and colleagues (2016) [9] found prevalence rates of 26.7% and 28.3% respectively in semi-structured diagnostic interviews. Looking only at current ADHD symptomatology, the prevalence rates of inattention, hyperactivity, and impulsivity were 23.3%, 12.5% and 21.7%, respectively [117]. Retrospective childhood ADHD was estimated at 35.2% [11] and 17.5% [117].

Five studies assessed ADHD in clinical populations of women with ED. High ADHD prevalence was found, especially among women with ED involving binging/purging behavior (AN-BP, EDNOS-BP, BN): from 10.2% to 49.8% [120,121,122,123,124]. However, Halevy-Yosef and colleagues (2019) [122] observed no significant difference in terms of ADHD prevalence between ED patients with BE (16.6%) and those without BE (13.6%) (*p* = 0.392).

After assessing disordered eating in a general non-clinical population, Brewerton & Duncan (2016) [118] found that the prevalence of ADHD was significantly higher in adults with lifetime or past 12-month disordered eating (BED, BN and binge eating), except for men diagnosed with lifetime disordered eating, and especially BED (see details Table 4). Similarly, in a sample of adults with major depressive or bipolar disorder, Woldeyohannes and colleagues (2015) [119] found an ADHD diagnosis rate of 20.8% among those with binge-eating behavior compared to 12.5% among those who did not binge (*p* = 0.018).

#### 3.2.3. ADHD and Disordered Eating

##### Children and Adolescents

Twelve studies explored the association between ADHD and addictive-like eating in children or adolescents.

Kim and colleagues (2018) [116] found that children with overeating had higher scores on the K-ARS (Korean version of the ADHD rating scale assessing ADHD symptom severity), increasing with frequency of overeating. Egbert and colleagues (2018) and Halevy-Yosef and colleagues (2019) conducted studies with individuals with clinical obesity and clinical ED respectively, and found that ADHD scale scores (Child Behavior Checklist, CBCL and ADHD-RS respectively) were higher in groups with dysregulated eating (56.17, Standard Deviation (SD) = 8.26 vs. 54.42, SD = 6.18, *p* < 0.05) [113] or binge eating (22.92, SD = 9.78 vs. 19.86, SD = 10.48, *p* < 0.001) [122]. In the clinical ED sample, further investigations found that severity of ADHD inattention symptoms was greater among binge-eating than non-binge eating individuals and controls (Bonferroni corrected *p* = 0.0003), and that severity of ADHD hyperactivity/impulsivity symptoms was greater in binge-eating and non-binge eating individuals than in controls (Bonferroni corrected *p* < 0.01). Patients who reported binging/purging behavior scored higher on both inattentive and hyperactivity/impulsivity ADHD subscales [122]. Kurz and colleagues (2017) [125] used a laboratory test meal and found no difference between individuals with ADHD and controls in loss of control overeating, liking for food and desire to eat.

Two studies conducted with non-clinical samples of children found that ADHD symptoms [126] and ADHD diagnosis [127] were related to emotional overeating. One of these studies [127] with 4-year-old children found a positive association between ADHD scale scores and eating behaviors, especially food responsiveness and emotional overeating. Moreover, children who scored in the medium and highest tertiles of the responsiveness scale and in the highest tertile of the emotional eating scale scored higher on the ADHD scales. In girls, food responsiveness was significantly associated only with impulsivity symptoms; in boys, it was significantly associated with inattentive and hyperactivity symptoms, while emotional overeating was significantly associated only with hyperactivity symptoms.

Some studies corroborated these results through correlation analysis. They found that ADHD severity was positively correlated with objective overeating (r = 0.10, *p* < 0.05), objective binge eating (r = 0.17, *p* < 0.01) [113], BN symptoms (r = 0.19, *p* < 0.0001), emotional overeating (r = 0.31, *p* < 0.0001) and emotional undereating (r = 0.28, *p* < 0.0001) [126], and with disordered eating as assessed on scales including the EAT-26 (ED severity, r = 0.53, *p* < 0.0001), EDE-Q (disordered eating behavior, r = 0.48, *p* < 0.0001), EDI-2 (impulse regulation and interoceptive awareness subscales, r = 0.65, *p* < 0.001 and r = 0.66, *p* < 0.001 respectively) [122].

Four studies conducted regression analyses and found a significant association between ADHD and disordered eating, and more specifically addictive-like eating behavior. These studies showed that ADHD symptoms were associated with loss of control overeating and binge eating [113], food preoccupation and oral control (i.e., self-control of eating and pressure from others to eat) [112]. Similarly, ADHD diagnosis was associated with loss of control overeating [105] and binge eating [103]. Egbert and colleagues (2018) [113] demonstrated that ADHD symptoms were positively associated with frequency of objective binge eating and objective overeating (respectively 6% and 5% increase in frequency of objective binge eating and objective overeating for every one-point increase in ADHD symptoms, ꭓ^2^(1) = 16.61, *p* < 0.001; ꭓ^2^(1) = 10.64, *p* < 0.01), but not subjective binge eating (ꭓ^2^(1) = 1.30, *p* = 0.25).

Further investigations involving mediation analyses highlighted the mediator role of loss of control overeating and binge eating in the relation between ADHD and BMI [103,105].

Four longitudinal studies found a positive association between ADHD symptoms during early-childhood and addictive-like eating behavior in later childhood or adolescence [128,129,130]. One of these studies [128] found a significant effect of ADHD symptoms on change in eating behaviors from early childhood (around 4 years old) to later childhood (around 7 years). They found that ADHD symptomatology was associated with changes in food responsiveness and emotional overeating when attention symptoms occurred, and only in emotional overeating when hyperactivity symptoms occurred. Conversely, the effect of eating behaviors on changes in ADHD symptomatology from early childhood to later childhood was not significant [128]. According to Sonneville and colleagues (2015) [130], mid- and late-childhood hyperactivity/impulsivity symptoms were correlated with mid- and late-childhood overeating and late-childhood BMI, leading to strong desire for food in early adolescence, correlated with binge eating in mid-adolescence. These results suggest that ADHD hyperactivity/impulsivity symptoms may lead indirectly to binge eating through overeating and desire for food. Similarly, Zhang and colleagues (2020) [131] found that ADHD symptoms at 14 predicted the development of binge eating (OR: 1.27, 95% CI: 1.03–1.57, *p* = 0.024) and purging (OR: 1.35, 95% CI: 1.12–1.64, *p* = 0.0016) behaviors at 16 or 19. However, Yilmaz and colleagues (2017) [129] found that only high inattention combined with high hyperactivity/impulsivity throughout childhood and adolescence predicted disordered eating, such as bulimia nervosa, in late adolescence (*p* < 0.01).

##### Adults

Thirteen studies focused on the association between ADHD and disordered eating in adults.

In a study with mood disorder outpatients, Woldeyoannes and colleagues (2015) [119] found no association between BE and childhood or adult ADHD (OR = 1.33, 95% CI: 0.40–4.49; OR = 1.05, 95% CI: 0.43–2.58 respectively). However, individuals with both BE and bipolar disorder had significantly higher scores on the WURS (retrospective childhood ADHD scale) and the ASRS (current adult ADHD scale; *p* = 0.007 and *p* < 0.001, respectively). Nazar (2018) [132] found no difference in binge eating between students with and without ADHD (*p* = 0.07), but greater binge eating among those with comorbid ADHD-ED (*p* < 0.001). In individuals with ADHD diagnosis, there was no difference between individuals with and without ED comorbidity in terms of inattentive and hyperactivity/impulsivity symptomatology (*p* = 0.53 and *p* = 0.75 respectively). Van der Oord and colleagues (2017) [133] assessed individuals with severe obesity and found that only comorbid BE was associated with an increase in ADHD symptomatology, mainly inattentive symptoms (*p* < 0.01). In this population, ADHD diagnosis was associated with bulimic symptoms, greater binge eating and higher FA scores [9,11]. Similar results were found when childhood ADHD was retrospectively assessed [11].

Six publications involved samples of individuals with ED. They found that ADHD symptomatology and diagnosis were associated with ED, especially binging/purging behaviors such as BN and AN binge/purge subtype, which were related to inattentive symptoms [122,123,124]. However, Halevy-Yosef and colleagues (2019) [122] found no differences in ASRS scores between ED with and without binging/purging behavior after Bonferroni correction. ED symptoms related to ADHD symptomatology were mostly addictive-like eating behaviors such as binge eating, purging and loss of control overeating [120,122]. Individuals diagnosed with ED scored higher on disordered eating scales if they also had ADHD. Ferre and colleagues (2017) [134] and Sala and colleagues (2018) [124] reported higher scores on the EAT-40 (assessing disordered eating) and BITE-symptomatology subscale (assessing binge eating symptomatology) among ED patients with than without comorbid ADHD symptomatology. However, while Ferre and colleagues (2017) [134] found similar results for binge-eating severity on the BITE-severity subscale, Sala and colleagues (2018) [124] found no significant difference between individuals with and without ADHD diagnosis. Carlucci and colleagues (2017) reported significant small multivariate effect of ED diagnosis on ASRS-total score (F(4992) = 2.43, *p* = 0.046), which was not found for either inattentive or hyperactivity-impulsivity factors (*p* = 0.06 and *p* = 0.016 respectively) [123]. Finally, a high baseline ASRS-total score (>18) was associated with a lower rate of ED recovery at 1 year follow-up (72.1% vs. 46.7%, *p* = 0.001), especially for binging (75.1% vs. 48.5%, *p* = 0.003), purging (74.0% vs. 47.6%, *p* = 0.001) and loss of control overeating (75.6% vs. 47.4%, *p* < 0.001) symptoms. This association remained significative only with ASRS inattentive factor, especially for binging and loss of control overeating. Regression analyses confirmed the predictive role of high ASRS scores on the persistence of disordered eating (OR = 2.59, 95% CI: 1.36–4.91) [121].

Among the six studies that analyzed the correlations between ADHD symptomatology and disordered eating, three were conducted with a student population and found positive correlations between ADHD and bulimic symptoms (r = 0.34, *p* < 0.001) [135] and binge eating ([132]: r = 0.43, *p* < 0.001; [136]: r = 0.21, *p* < 0.001). Similar results were found for patients with ED [122,123] or severe obesity [117], for both inattentive (r = 0.33–0.36, *p* < 0.001) and hyperactivity/impulsivity symptoms (r = 0.22–0.30, *p* < 0.001). However, Hanson and colleagues (2019) [136] found no correlation between binge eating and ADHD-Inattentive symptoms for men in their student sample (r = 0.19, *p* > 0.05).

Five studies conducted regression analyses. Woldeyoannes and colleagues (2015) [119] showed that correlates of BE reported by patients with mood disorder did not include symptomatology of current ADHD or retrospectively assessed childhood ADHD (adjusted Odds Ratio (aOR) = 1.33, 95% CI: 0.40–4.49, aOR = 1.05, 95% CI: 0.43–2.58 respectively). However, the other four studies (with students, patients with severe obesity or with ED) found a significant association between ADHD symptoms/diagnosis and addictive-like eating behavior such as binge eating [11,132,136], disordered eating, bulimic symptoms [134] and FA [11]. Ferre and colleagues (2017) [134] found that patients with ED and ADHD symptoms scored higher on the EAT-40 (assessing disordered eating), the BITE-symptomatology sub-scale (assessing binge eating symptomatology) and BITE-severity sub-scale (assessing binge eating severity). The predictive power of ADHD symptoms on these scales was 14%, 7% and 11% respectively.

Nielsen and colleagues (2017) and Brunault and colleagues (2019) reported that addictive-like eating was more strongly associated with adulthood than childhood ADHD ([117]: the correlation between ADHD symptoms and ED psychopathology scales was stronger for adulthood than childhood ADHD symptoms; [11]: ORs for the association between ADHD symptoms and FA or binge eating were higher for adulthood than childhood ADHD symptoms).

### 3.3. Indirect Association between ADHD and Disordered Eating through Negative Affectivity and Disrupted Emotion Self-Regulation

#### 3.3.1. ADHD, Negative Affectivity, and Disrupted Emotion Self-Regulation

##### Children and Adolescents

Two studies conducted with children found that ADHD group had more adolescent with clinical internalizing (i.e., Strengths and Difficulties Questionnaire subscale investigating emotional symptoms and peer problems) (33.3% vs. 16.0%; OR: 2.6; 95% CI: 1.9–3.7) [106] and ADHD symptoms was significatively correlated with depressive symptoms (r = 0.49, *p* < 0.0001) [126].

##### Adults

Among the studies included in this review, twelve focused on the comorbidity of ADHD symptoms and negative affectivity. Many of them identified a high correlation between ADHD symptoms and anxiety (rated from 0.28, *p* < 0.008 to 0.42, *p* < 0.001) [9,120] and depressive symptoms (rated from 0.29, *p* < 0.001 to 0.38, *p* < 0.001) [9,120,132]. Both inattention and hyperactivity/impulsivity symptoms were correlated with anxiety (r = 0.68, *p* < 0.0001 and r = 0.57, *p* < 0.0001 respectively) and depressive symptoms (r = 0.56–0.63, *p* < 0.001 and r = 0.41–0.51, *p* < 0.001) [117,122]. As reported by several publications [109,110,111], Jacob and colleagues (2018) [108] showed that individuals who screened positive for adult ADHD (ASRS) had a greater risk for anxiety disorder (33.6% vs. 5.1%, *p* < 0.001), mood disorders such as major depressive disorder (17.1% vs. 2.1%, *p* < 0.001), as well as borderline personality disorder traits (24.0% vs. 2.7% *p* < 0.001). Gorlin and colleagues (2016) [110] did not find an association between ADHD diagnosis and higher anxiety and depressive disorders. However, that study was conducted with psychiatry outpatients who may have been under medication for mood and anxiety disorders.

ADHD symptomatology was also associated with a higher number of stressful life events (3 vs. 1.7 *p* < 0.001) and more frequent perceived stress (85.9% vs. 59.1%, *p* < 0.001) [108]. In addition, ED patients with ADHD symptoms had higher anxiety (*p* = 0.02) [124], higher perceived stress and lower life satisfaction and perceived social support than those with ADHD symptoms [134]. These results indicate high rates of negative affectivity for ADHD individuals. Both inattentive and hyperactivity-impulsivity symptoms were shown to be correlated negatively with effortful control-regulative temperament (inattention: r = −0.556, *p* < 0.001, hyperactivity: r = −0.348, *p* < 0.001 and impulsivity: r = −0.476, *p* < 0.001) [117], and positively with emotion regulation difficulties (r = 0.42, *p* < 0.001 for both inattentive and hyperactivity/impulsivity ADHD symptoms) [135].

#### 3.3.2. Negative Affectivity and Disrupted Emotion Self-Regulation as Mediators in the Association between ADHD and Disordered Eating

##### Children and Adolescents

Tong and colleagues (2017) [126] clarified the association between ADHD symptoms and addictive-like eating behavior by introducing a potential mediating effect of depression in this relationship. Their data are in line with the hypothesis that ADHD is associated with bulimia and emotional overeating through depression. Koch and colleagues (2020) [137], who investigated the incidence of mental disorders in zero to three-year-old children, suggested that the associations between emotional and affective disorders and ED and ADHD respectively were stronger than the direct association between feeding and eating disorders and ADHD. Indeed, the comorbidity between feeding and eating disorders and ADHD was OR = 15.4 (95% CI 9.6–24.7), whereas the comorbidity between EAD (i.e., emotional and affective disorders) and feeding and eating disorders was OR = 66.8 (95% CI 42.6–104.7) and between EAD and ADHD was OR = 150.7 (95% CI 95.1–238.7). In a sample of overweight or obese children, Gowey and colleagues (2017) [112] found that negative affectivity mediated the relationship between ADHD symptoms and disordered eating. They found significant interactions between body dissatisfaction and both inattentive and hyperactivity/impulsivity ADHD symptoms with an effect on addictive-like eating behavior, especially food preoccupation and oral control.

##### Adults

A study with adults by Jacob and colleagues (2018) [108] reported a relationship between ADHD symptoms and possible ED, especially uncontrolled eating symptoms largely explained by anxiety disorder (40% for possible ED, 33% for uncontrolled eating) and stressful life events (28% for possible ED, 24% for uncontrolled eating). Another study found that the odds ratio of ADHD-ED association was considerably attenuated after adjusting for comorbid psychiatric disorders (such as mood and anxiety disorders), especially for BN (before adjusting for psychiatric disorders: OR: 28.24, 95% CI: 6.33–126.01; after adjusting for psychiatric disorders: OR: 5.04, 95% CI: 1.15–22.08) [109].

Similarly, Kaisari and colleagues (2018) [138] found that ADHD inattentive and hyperactivity/impulsivity symptoms were both directly and indirectly associated with binge eating through negative affectivity (anxiety, depression and perceived stress). Moreover, after controlling for depressive and anxiety symptoms, there was no longer a correlation between ADHD symptoms and BMI (inattention: r = −0.031; *p* = 0.350 and hyperactivity/impulsivity: r = −0.05; *p* = 0.307 respectively) [9].

Christian and colleagues (2020) [135] found that negative urgency and emotion self-regulation difficulties were associated with both bulimic and ADHD symptoms, highlighting a possible shared pathway to both ADHD and ED symptoms. Further investigations revealed an impact of negative urgency and emotion self-regulation difficulties in the association between ADHD and ED, especially bulimic symptoms. These results support the hypothesis that negative urgency and emotion dysregulation mediate the association between ADHD and disordered eating.

Williamson and colleagues (2017) [139] investigated the role of emotion self-regulation and ADHD symptoms in the weight loss of obesity patients after bariatric surgery. The interaction between ADHD symptomatology and emotion self-regulation accounted for 13% of the weight loss variance. The results also indicated an inverse association between ADHD symptoms and weight loss 12 months post-surgery among patients with low scores on emotion self-regulation (36.7% of the sample).

## 4. Discussion

The purpose of the present study was to investigate the association between ADHD symptomatology, disordered eating, especially addictive-like eating behavior, and emotion self-regulation. We noted a significant association with disordered eating (especially addictive-like eating behavior) in 38 publications, eight of them highlighting the mediator role of negative affectivity and emotion dysregulation. This trend was qualified in 19 publications; 16 publications reported differences depending on type of disordered eating behavior, gender or ADHD symptoms. The majority of results thus suggest that both childhood and adulthood ADHD symptomatology is associated with a higher risk of addictive-like eating behavior, especially binging and/or purging, loss of control overeating, emotional overeating and binge eating, bulimic symptoms, as well as a strong desire for food, food responsiveness and food preoccupation. Furthermore, some authors suggest that ADHD symptoms during early childhood lead to disordered eating during later childhood or adolescence.

Several authors found that severe obesity or ED comorbidities increased the strength of the association between ADHD and disordered eating, especially binge eating. Their results indicate that binge eating and purging behavior play a key role in this association, particularly the BN and AN binge/purging subtype. According to Granero and colleagues (2014) [19], this subtype has the highest rate of FA, supporting the hypothesis of a strong association between ADHD and FA. Other publications show that ADHD psychostimulant treatment can improve ED symptoms, suggesting that ADHD and disordered eating share pathways [131,140,141]. According to Zhang and colleagues (2020) [131], low grey matter volume in the orbitofrontal cortex is a mediator between ADHD symptoms and the development of purging, binging/purging behaviors and depression. Moreover, dopaminergic reward pathways are implicated in both ADHD and disordered eating. In ADHD, disruption of the dopaminergic system involves impulse control deficits, inattention and reward sensitivity. These features increase the risk of resorting to food, and even of FA, with palatable food seen as a natural reward [140].

Longitudinal studies demonstrate that a combination of high inattention and hyperactivity/impulsivity symptoms in childhood lead to increasing BMI in late childhood and to ED in adolescence through addictive-like eating behaviors. However, some publications reported that disordered eating is particularly linked to inattentive symptoms. It is not possible in this systematic review to draw clear conclusions about the involvement of inattentive and/or hyperactivity/impulsivity ADHD symptoms in the association between ADHD and addictive-like eating behavior, and further investigations are needed.

The second aim of this systematic review was to examine the mediator role of negative affectivity and emotion self-regulation in the association between ADHD and addictive-like eating behavior. We showed that high ADHD severity would be associated with a high risk of disrupted emotion regulation, negative affectivity (comorbid anxiety and mood disorders, and perceived stress), which mediate the link between ADHD symptomatology and disordered eating, especially addictive-like eating behavior. Some studies show that ADHD symptoms are associated with high emotion dysregulation [117,135], impacting the ability to cope with daily difficulties, and involving greater negative affectivity and a higher risk of mood disorder comorbidity. As expected, some studies indicated that negative affectivity and emotion dysregulation mediates the association between ADHD and addictive-like eating behavior [108,109,112,126,135,137,138,139], supported by publications which showed association between ADHD and emotional eating [126,127,128]. Negative affectivity and lack of emotion regulation, commonly observed in ADHD, would trigger food intake. Results also suggest that individuals with ADHD tend to act rashly when experiencing negative affectivity (negative urgency), which is associated with disordered eating, such as binging [135].

The studies included in this systematic review suggest a pattern of links between ADHD symptomatology, negative affectivity, emotion regulation, and addictive-like eating behaviors (Figure 3). ADHD symptomatology would lead to greater difficulty coping with daily life, due to emotion dysregulation. Due to their inability to regulate negative affectivity, people with ADHD tend to run away from them by seeking positive sensations such as eating. Impulsivity and negative urgency would further encourage disordered eating behaviors such as binge eating, leading to greater BMI. The urge to eat when in a negative affectivity indicates an addictive process involving similar dopaminergic pathways to ADHD.

A better understanding of the mechanisms underlying the association between ADHD symptomatology and disordered eating suggests new approaches to psychological interventions. In view of the high incidence of disordered eating among people with ADHD, it seems important to identify any maladaptive eating behavior. Interventions aimed at assessing and targeting emotion dysregulation could be an appropriate way of preventing disordered eating behavior and FA, as well as comorbid anxiety and depression disorders. Integrative cognitive-affective therapy (ICAT) adapted to BN and BED targeting emotion regulation (identification of emotional states, especially negative ones, self-monitoring of eating patterns, behaviors and emotions) has been shown to be effective in reducing the frequency of binge eating [142]. Similarly, early detection of ADHD symptoms among people with disordered eating would enable suitable intervention programs to be set up, particularly to treat poor impulse control and emotion dysregulation. A number of personal characteristics that have a negative impact on ED therapy outcome should be identified, including the presence of ADHD symptoms. ADHD symptomology could be a predictor of the outcome of bariatric surgery in individuals with severe obesity [139]. It is thus essential to identify inattention and hyperactivity/impulsivity symptoms in order to provide appropriate joint interventions. For example, Cortese and colleagues (2007) advocated a dual intervention of medication (to reduce comorbid ADHD and ED symptomatology) and cognitive behavioral therapy (to control impulsive and maladaptive behavior, and emotion regulation) [123,143].

This review has a number of limitations. First, it does not provide any causal link. Indeed, as far as we know, no study investigated the effect of ADHD negative affectivity or emotion dysregulation therapeutic interventions on addictive like eating behavior. This link could be of interest for further studies. Moreover, this systematic review includes only qualitative and no quantitative analyses. The variety of populations studied (individuals diagnosed with ADHD, different types of disordered eating, severe obesity, students, etc.) and methods used to assess ADHD and disordered eating make it difficult to draw clear conclusions. In addition, some studies were based on ADHD diagnosis criteria of the DSM-IV-TR and others on DSM5 criteria, with a change of symptom onset from 7 to 12 years of age, making it difficult to compare results. Another limitation involves publications which did not provide necessary information about current medication. Indeed, medication can conceal symptoms of disrupted emotion and eating, so there is an impact on results of investigations. Furthermore, as only a few studies assessed food addiction directly, we included those involving addictive-like eating symptoms and various aspects of food addiction. It should be noted that the addictive nature of food is still under debate, notably whether features of substance addiction can be applied to food, the addictive power of palatable food, common features such as tolerance and withdrawal, and the distinction between food addiction and binge eating. However, people presenting with this type of pathological eating suffer in similar ways as those with substance use disorder, including “feelings of deprivation when the substance is withheld, a propensity to relapse during periods of abstinence, and consumption that persists despite awareness of negative health, social, financial, or other consequences” [144]. The publications reviewed have their own limitations. According to the DSM-5, childhood ADHD symptoms are used to diagnose adult ADHD. However, several studies involving adult ADHD did not investigate childhood symptoms. Some studies only used self-administered questionnaires to assess disordered eating and ADHD. This type of assessment is not as efficient as an interview with a clinician.

Future studies should investigate in greater depth emotion regulation difficulties in comorbid adult ADHD and addictive-like eating behavior, and the involvement of specific ADHD symptoms such as inattention, impulsivity and hyperactivity. This could clarify which emotion regulation strategies and ADHD symptoms should be targeted in clinical interventions. It would be interesting to investigate specific symptoms of ED in order to identify common sub-groups. The majority of studies of ADHD symptomatology in people with disordered eating were conducted with female samples, although some authors noted male-female differences in the relationship between ADHD and disordered eating. Future studies should thus investigate distinctive male characteristics in order to determine whether clinical interventions should be gender-specific. In addition, in order to identify causal links between ADHD symptomatology and addictive-like eating behavior, more longitudinal studies are needed. This would make it possible to set up early interventions with children with ADHD and investigate the impact on ADHD symptomatology, eating behaviors and risk of obesity in adolescence and adulthood. An important area of research would be to focus on the interplay between dysregulation of sleep, weight gain and emotional dysregulation, as it has been suggested by some authors [145] that alterations in sleep/arousal may be related to ADHD and weight gain/disordered eating and sleep deprivation may exacerbate emotional dysregulation [146].

## 5. Conclusions

Despite its limitations, this review provides information about the co-occurrence of ADHD symptoms and addictive-like eating behavior. It confirms the strong association between ADHD, emotion dysregulation and binge eating/addictive-like eating behavior in both clinical (i.e., people with ED or ADHD) and non-clinical populations. The data support the hypothesis of a mediating role of negative affectivity and emotion self-regulation difficulties in the association between addictive-like eating behavior and ADHD. This review paves the way for future therapeutic interventions that could improve clinical outcomes for people with ADHD and disordered eating.

## Figures and Tables

**Figure 1 nutrients-12-03292-f001:**
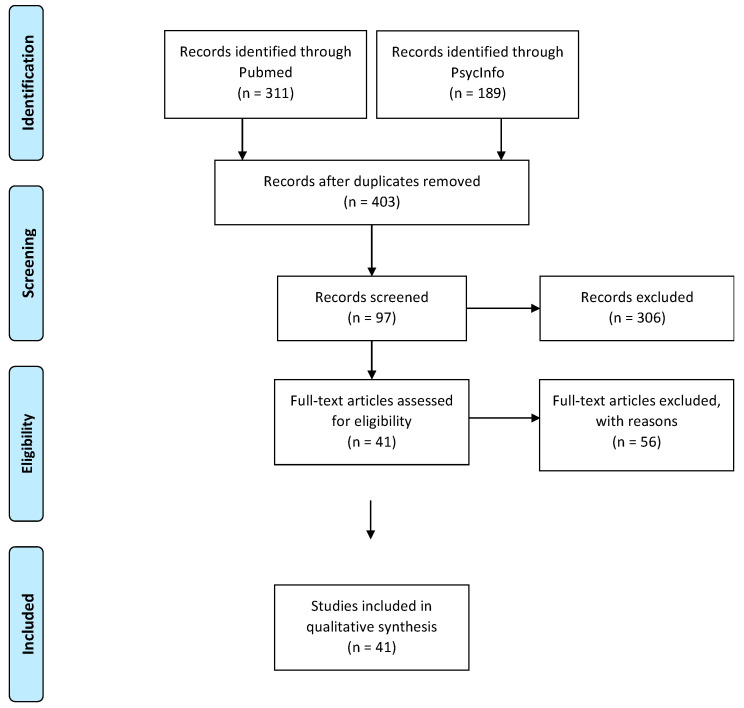
Study selection flow chart.

**Figure 2 nutrients-12-03292-f002:**
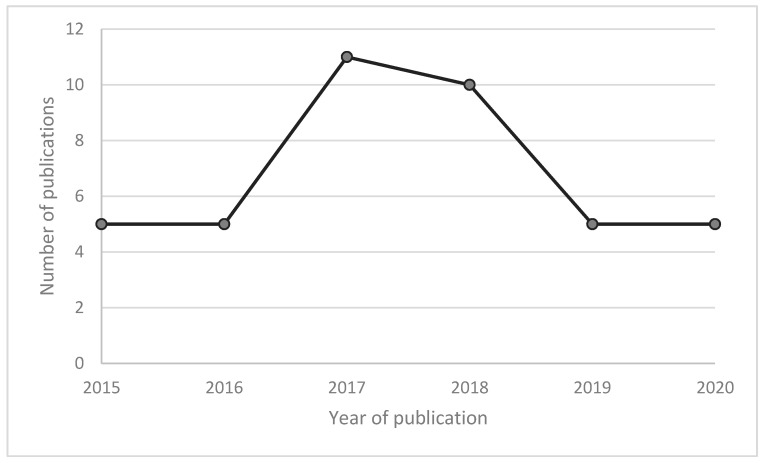
Number of publications from 2015 to 2020.

**Figure 3 nutrients-12-03292-f003:**
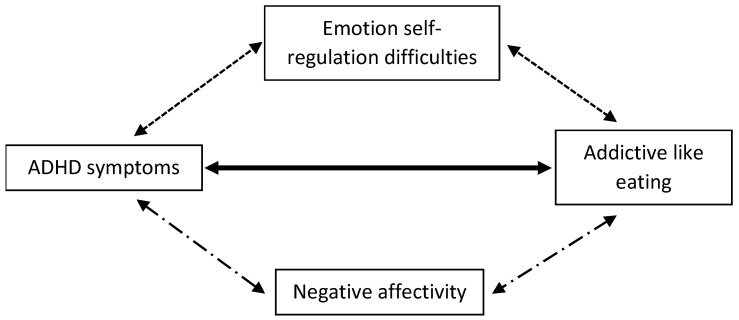
Model illustrating association between ADHD symptoms and disordered eating mediated by emotion self-regulation difficulties and negative affectivity. 
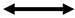
: [9,11,103,105,112,113,116,117,120,121,122,123,124,126,127,129,130,131,132,133,134,135,136,137,138]. 
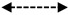
: [135]. 
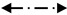
: [108,109,112,126,137,138].

**Table 1 nutrients-12-03292-t001:** Inclusion and exclusion criteria.

Inclusion Criteria		Exclusion Criteria	
**I1**	Key words cited in the abstract	**E1**	Key words not cited in the title/abstract
**I2**	Date of publication: January 2015 to June 2020	**E2**	Publication before January 2015
**I3**	Journal article with peer-review	**E3**	Book chapter, letter to the editor or other non-empirical type of publications
**I4**	Written in English or French	**E4**	Paper not written in English or French
**I5**	Empirical research	**E5**	Review and meta-analysis papers
**I6**	Focus on the association between ADHD and eating behavior	**E6**	Focus on treatment, medical imaging, genetics
**I7**	ADHD and disordered eating symptoms in the same individual	**E7**	Focus on the impact of parents’ disordered eating or BMI on their child’s ADHD symptoms
**I8**	Assessment of ADHD and disordered eating symptoms		

Note: ADHD: Attention-Deficit Hyperactivity Disorder; BMI: Body Mass Index.

**Table 2 nutrients-12-03292-t002:** Disordered eating prevalence among individuals with Attention-Deficit Hyperactivity Disorder symptomatology.

Population	Country	Children- Adolescents	Adults	ADHD Diagnosis Instruments	N	Age Mean (SD) (Years)	Gender Female (%)	Disordered Eating	ADHD Symptoms		Non-ADHD Symptoms		Statistics		
									*n*	Disordered Eating Prevalence (%)	*n*	Disordered Eating Prevalence (%)	Odds Ratio	95% Confidence Intervals	*p*
**General population**															
[105]	USA	x		K-SADS PL and CRPS	79	11.0 (1.9)	48.1	LOC-E	31	70.5	7	20	12.68	3.11–51.64 ^a^	
[106]	Australia	x		SDQ or parent-reported ADHD diagnosis or medication	2672	14.9 (0.3)	0	Regular objective BE		2.9		0.3	9.4	1.7–52.8	
								Partial syndrome BN		1.6		1.5	1.0	0.1–8.0	
								Partial syndrome BED		1.3		0.2	6.2	0.6–61.1	
							100	Regular objective BE		6.5		2.2	3.1	0.7–14.0	
								Partial syndrome BN		6.5		3.6	1.9	0.4–8.2	
								Partial syndrome BED		**0**		0.6	-	-	
[107]	Sweden		x	DSM-IV criteria	18,029	33.6 (7.6)	55.6	BE behavior	113	7.17		-	3.65	2.72–4.91	
								DSM-5 BE behavior	58	3.72		-	3.01	2.09–4.35	
								DSM-5 BED	7	0.45		-	2.55	1.11–5.86	
								DSM-5 BN	48	3.11		-	3.09	2.09–4.56	
[108]	UK		x	ASRS	7403	46.3 (18.6)	51.4	Self-sick for feeling full		8.5		2.7	2.79	1.76–4.42 ^b^	
													1.26	0.63–2.51 ^c^	
								Uncontrolled eating		22.8		6.4	3.94	2.94–5.28 ^b^	
													1.67	1.14–2.46 ^c^	
								Possible ED		19.2		5.7	3.48	2.56–4.72 ^b^	
													1.32	0.82–2.13 ^c^	
[13]	France		x	WURS-25 + ASRS	1517	20.6 (3.6)	68.2	Food addiction	12	14.1	57	4.0	2.27	1.05–4.88 ^d^	
								Any ED	28	32.9	249	17.4	1.33	0.76–2.33 ^d^	
[109]	USA		x	DIS-IV (childhood) + adult ACDS	4719	31 (DNS)	52.1	Past 12-month any ED					9.74	4.23–22.40 ^b^	
													2.84	1.22–6.63 ^c^	
								Past 12-month BED					4.53	1.82–11.24 ^b^	
													1.65	0.67–4.04 ^c^	
								Past 12-month BN					28.24	6.33–126.01 ^b^	
													5.04	1.15–22.08 ^c^	
								Past 12-month subthreshold BED					5.55	1.90–16.24 ^b^	
													3.83	0.94–15.67 ^c^	
**Psychiatry outpatients**															
[103]	USA	x		DSM-IV criteria	252	10.8 (3.7)	47.2	BE	28	26	3	2.0	-	-	***
[110]	USA		x	SCID-IV	1134	39.7 (14.4)	58	Any ED	19	9.3	35	3.8	2.67	1.45–4.80	
**Patients with obesity**															
[104]	Sweden	x		Medical records or DAWBA	40	12.4 (3.0)	48.7	LOC-E	5	55.6	21	67.7	-	-	>0.05
[11]	France		x	DIVA 2.0	105	46.5 (10.7)	86.7	Food addiction	8	28.6	7	9.1	4.00	1.29–12.40	
								Significant distress in relation to food	9	32.1	9	11.7	3.58	1.25–10.30	
**ADHD outpatients**															
[111]	France		x	Children-MINI adapted for adults	81	34.8 (11.6)	37	Bulimia nervosa	7	8.6	-	-	-	-	-
[29]	Norway		x	DSM-IV criteria	533	36.2 (11.3)	100	Any ED	36	13.0	-	-	-	-	-
						37.4 (10.7)	0	Any ED	3	1.1	-	-	-	-	-

Note: N or n: group size; SD: Standard Deviation; ADHD: Attention-Deficit Hyperactivity Disorder; DSM-IV: Diagnostic and Statistical Manual of mental disorders, fouth edition; LOC-E: Loss Of Control overEating; BE: Binge Eating; BED: Binge Eating Disorder; BN: Bulimia Nervosa; ED: Eating Disorder; K-SADS PL: Schedule for Affective Disorders and Schizophrenia for school-age children-Present and Lifetime Version; CRPS: Conners-3 Parent Rating Scale-Revised; SDQ: Strengths and Difficulties Questionnaire; ASRS: Adult ADHD Self-Report Scale; WURS: Wender Utah Rating Scale; DIS-IV: Diagnostic Interview Schedule for DSM-IV; ACDS: ADHD Clinical Diagnostic Scale; SCID-IV: The Structured Clinical Interview for DSM-IV; DAWBA: Development and Well-Being Assessment; DIVA: Diagnostische Interview Voor ADHD; MINI: Mini International Neuropsychiatric Interview; DNS: data not specified; ^a^: model adjusted for age, sex, race, body mass index z score; ^b^: model adjusted for age, race, sex; ^c^: model adjusted for age, race, sex and lifetime diagnosis of psychiatric comorbidities; ^d^: model adjusted on universities (place of recruitment), cursus and financial difficulties; *** *p* < 0.001.

**Table 3 nutrients-12-03292-t003:** Attention-Deficit Hyperactivity Disorder prevalence among population with overweight or obesity.

Population	Country	Children-Adolescents	Adults	N	Age Mean (SD) (Years)	Gender Female (%)	Mean BMI or zBMI (SD)	ADHD Instruments	Childhood ADHD Prevalence (%)	Adult ADHD Prevalence
**Non clinical population**										
[112]	USA	x		220	10.3 (1.4)	53.6	2.19 (0.38)	CBCL	5.0	
**Clinical obesity population**										
[113]	USA	x		385	10.9 (2.3)	63	2.26 (0.35)	CBCL	11.0	
[104]	Sweden	x		76	12.4 (3.0)	48.7	3.40 (0.50)	Medical records or DAWBA	18.4	
[9]	Brazil		x	106	39.0 (10.7)	100	39.21 (5.29)	K-SADS adapted for adults, DSM IV		28.3 ^a^
[117]	Germany		x	120	41.0 (11.5)	79.2	47.76 (7.41)	WURS-k + CAARS-S:S	17.5 ^b^	8.3 ^a^
[11]	France		x	105	46.4 (10.7)	86.7	46.90 (7.80)	DIVA 2.0	35.2 ^b^	26.7 ^a^

Note: N: group size; SD: Standard Deviation; BMI: Body Mass Index; zBMI: Body Mass Index z score; ADHD: Attention-Deficit Hyperactivity Disorder; CBCL: Child Behavior Checklist; DAWBA: Development and Well-Being Assessment; K-SADS: Schedule for Affective Disorders and Schizophrenia for school-age children; WURS-k: Wender Utah Rating Scale Short Version; CAARS-S:S: Conners’ Adult ADHD Rating Scale-Self-Report: Short Version; DIVA: Diagnostische Interview Voor ADHD; DSM IV: Diagnostic and Statistical Manual of mental disorders, Fourth edition ^a^: ADHD symptomatology since childhood as expected by DSM criteria; ^b^: retrospectively estimated.

**Table 4 nutrients-12-03292-t004:** Attention-Deficit Hyperactivity Disorder prevalence among disordered eating.

Population	Country	Children- Adolescents	Adults	ADHD Instruments	N	Mean Age (SD) (Years)	Gender Female (%)	Disordered Eating	Disordered Eating		Non Disordered Eating		Statistics		
									*n*	ADHD Symptoms Prevalence (%)	*n*	ADHD Symptoms Prevalence (%)	Odds Ratio	95% Confidence Intervals	*p*
**General population**															
[114]	Spain	x		K-SADS	962	DNS (12–16)	47.8	ED	11	31.4	80	8.4	5.03	2.37–10.64	
[116]	Korea	x		K-ARS	16,831	9.29 (1.71)	50.2	Every day overeating	68	21.1	-	-			
[115]	Iran	x		K-SADS PL	27,111	DNS (6–18)	48.6	Lifetime ED	-	7.6	-	3.9			0.026
[118]	USA		x	CIDI	1686	DNS (18–44)	100	Lifetime any ED	18	21.9	75	5.7	4.51	2.01–10.15 ^b^	
								Past 12-month any ED	10	30.6	83	6.1	7.11	2.61–19.39 ^b^	
								Lifetime BED	8	17.1	85	6.3	3.01	1.14–7.95 ^b^	
								Past 12-month BED	4	19.3	89	6.5	3.57	1.06–12.09 ^b^	
								Lifetime BN	10	33.2	83	6.1	7.93	2.75–22.85 ^b^	
								Past 12-month BN	6	56.7	87	6.3	21.15	3.76–118-98 ^b^	
								Lifetime any binge	16	18.7	77	5.9	3.66	1.71–7.87 ^b^	
								Past 12-month any binge	8	19.4	85	6.3	3.71	1.68–8.20 ^b^	
							0	Lifetime any ED	6	21.3	85	9.7	2.23	0.81–6.13 ^b^	
								Past 12-month any ED	4	45.9	87	9.7	6.48	1.33–31.60 ^b^	
								Lifetime BED	6	25.4	85	9.6	2.93	0.98–8.76 ^b^	
								Past 12-month BED	4	45.9	87	9.7	6.47	1.33–31.61 ^b^	
								Lifetime BN	1	66.9	90	9.9	18.18	1.39–238.40 ^b^	
								Lifetime any binge	11	19.4	80	9.5	2.39	1.17–4.91 ^b^	
								Past 12-month any binge	7	38.9	84	9.5	5.02	1.90–13.28 ^b^	
**Mood disorder outpatients ^a^**															
[119]	Canada		x	MINI	631	37.8–40.0 (12.0–12.4)	59.0	BE	26	20.8	61	12.5			0.018
**ED patients**															
[120]	Sweden		x	ASRS	1094	27.7 (8.7)	100	Any ED	346	31.6	-	-			
								BED	25	27.5	-	-			
								BN	156	37.1	-	-			
								AN-BP	13	35.1	-	-			
								AN-R	12	17.6	-	-			
								EDNOS-BP	102	31.0	-	-			
								EDNOS-R	38	25.7	-	-			
[121]	Sweden		x	ASRS	443	27.5 (8.5)	100	Any ED	45	10.2	-	-			
[122]	Israel	x	x	K-SADS PL	168	DNS (15–28)	100	BE	-	16.6	-	13.6			0.392
								BN	-	12.0	-	-			
								AN-BP	-	28.0	-	-			
								AN-R	-	9.0	-	-			
[123]	Canada		x	ASRS	500	27.6 (10.6)	95.2	Any ED	-	49.8					
[124]	France		x	WURS + BAADS	73	28.1 (7.3)	100	AN-R	3	8.1	-	-			
								AN-BP	9	32.1	-	-			
								BN	1	12.5	-	-			

Note: N: group size; SD: Standard Deviation; ADHD: Attention-Deficit Hyperactivity Disorder; K-SADS PL: Schedule for Affective Disorders and Schizophrenia for school-age children-Present and Lifetime Version; K-ARS: Korean version of the ADHD rating scale; CIDI: Composite International Diagnostic Interview; MINI: Mini International Neuropsychiatric Interview; ASRS: Adult ADHD Self-Report Scale; WURS: Wender Utah Rating Scale; BAADS: Brown Attention Deficit Disorder Scale; ED: eating disorder; BE: binge eating; BN: bulimia nervosa; BED: Binge Eating Disorder; AN: Anorexia Nervosa; BP: Binging/purging type; R: restrictive type; EDNOS: Eating Disorders Not Otherwise Specified; DNS: data not specified; ^a^: major depressive disorder or bipolar disorder; ^b^: logistic regression models adjusted for age and race/ethnicity.

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
