# Peer review of "Negative Affectivity and Emotion Dysregulation as Mediators between ADHD and Disordered Eating: A Systematic Review"

_nutrients, 2020, doi:10.3390/nu12113292_

Round 1
Reviewer 1 Report
Your article covers an interesting topic. You say that the data support the hypothesis of a mediating role of negative affectivity and emotion self-regulation difficulties in the association between addictive like eating behavior and ADHD. To really show that these variables are true mediators with causal links, one should study the effect on eating behaviour by treating and thereby diminishing negative affectivity and emotion self-regulation difficulties. If this would lead to less problems with addictive eating behavior it would give strong support to the hypothesis of a mediating role for these variables, but this has not been shown yet. I think you should mention this in your conclusion.
Author Response
Thank you for your interesting comment. You actually pointed out an important element that is missing in the literature. So, we added these information as part of the “limits” in the discussion: lines 681-684, “it does not provide any causal link. Indeed, as far as we know, no study investigated the effect of ADHD negative affectivity or emotion dysregulation therapeutic interventions on addictive like eating behavior. This link could be of interest for further studies.”

Reviewer 2 Report
This is an interesting and well written systematic review . The introduction covers the literature very well and exposes the scientific problem comprehensively. The Methods section describes the scientific approach appropriately. The results section is suitable as well as illuminating, and the results are well discussed. Interestingly, despite the limitation of including only qualitative analyzes and no quantitative analyzes, this review provides insight into the co-occurrence of ADHD symptoms and addictive eating behavior. Confirming the strong association between ADHD, emotional dysregulation, and binge eating / addictive eating behavior in both clinical and non-clinical populations. This review has the merit of driving research towards future therapeutic interventions that could improve clinical outcomes for people with ADHD and eating disorders.
I have no suggestions for improving the systematic review".
Author Response
Thank you for your comments. We are delighted to know that you appreciated our manuscript.
